# Enhanced fetal hematopoiesis in response to symptomatic SARS-CoV-2 infection during pregnancy

Mansour Alkobtawi[1,11], Qui Trung Ngô[1,11], Nicolas Chapuis[2,3], Romain H. Fontaine[1], Mira El Khoury [4], Matthieu Tihy[5], Nawa Hachem[4], Aude Jary [6], Vincent Calvez[6], Michaela Fontenay[2,3], Vassilis Tsatsaris[7,8,9,12], Sélim Aractingi [1,10,12 ✉] & Bénédicte Oulès [1,10,12]

**Abstract**

**Background** Pregnant women and their fetuses are particularly susceptible to respiratory pathogens. How they respond to SARS-CoV-2 infection is still under investigation.

**Methods** We studied the transcriptome and phenotype of umbilical cord blood cells in pregnant women infected or not with SARS-CoV-2.

**Results** Here we show that symptomatic maternal COVID-19 is associated with a transcriptional erythroid cell signature as compared with asymptomatic and uninfected mothers. We observe an expansion of fetal hematopoietic multipotent progenitors skewed towards erythroid differentiation that display increased clonogenicity. There was no difference in inflammatory cytokines levels in the cord blood upon maternal SARS-CoV-2 infection. Interestingly, we show an activation of hypoxia pathway in cord blood cells from symptomatic COVID-19 mothers, suggesting that maternal hypoxia may be triggering this fetal stress hematopoiesis.

**Conclusions** Overall, these results show a fetal hematopoietic response to symptomatic COVID-19 in pregnant mothers in the absence of vertically transmitted SARS-CoV-2 infection which is likely to be a mechanism of fetal adaptation to the maternal infection and reduced oxygen supply.

**Plain language summary**

During pregnancy, women are more prone to respiratory infectious diseases. It is not known if COVID-19 infection has an adverse effect on the growing fetus. Here, we aimed to identify any potential effects of COVID-19 infection on the fetus by taking measurements from the umbilical cord blood cells. In mothers who displayed symptomatic COVID-19 infection, we observed an increased production of hematopoietic progenitor cells, especially the ones that are responsible for producing red blood cells. We think this might be a coping mechanism for the fetus, as the mother's body deals with the infection. Therefore, our work shows that growing fetuses do respond to maternal COVID-19 symptoms, even when they are protected in the womb from the infection and may never get infected by the mother.

[1] Université Paris Cité, CNRS UMR 8104, INSERM U1016, Institut Cochin, Cutaneous Biology Lab, Paris, France. [2] Université Paris Cité, CNRS UMR 8104, INSERM U1016, Institut Cochin, Normal and Pathological Hematopoiesis Lab, Paris, France. [3] Laboratory of Hematology, Hôpital Cochin, AP-HP.Centre-Université Paris Cité, Paris, France. [4] Sorbonne University, INSERM UMR-S 938, Saint-Antoine Research Center, CRSA, AP-HP, Saint-Antoine Hospital, Paris, France. [5] Department of Pathology, Hôpitaux Universitaires Genève, Genève, Switzerland. [6] Sorbonne Université, INSERM, Institut Pierre Louis d'Epidémiologie et de Santé Publique (iPLESP), AP-HP, Hôpital Pitié-Salpêtrière, Service de Virologie, Paris, France. [7] Department of Obstetrics, Maternité Port Royal, Hôpital Cochin, AP-HP.Centre-Université Paris Cité, Paris, France. [8] FHU PREMA, Paris, France. [9] Université Paris Cité, INSERM U1139, Pathophysiology & Pharmacotoxicology of the Human Placenta, Paris, France. [10] Department of Dermatology, Hôpital Cochin, AP-HP.Centre-Université Paris Cité, Paris, France. [11] These authors contributed equally: Mansour Alkobtawi, Qui Trung Ngô. [12] These authors jointly supervised this work: Vassilis Tsatsaris, Sélim Aractingi, Bénédicte Oulès. ✉email: selim.aractingi@gmail.com

Since 2020, SARS-CoV-2 infection has become a global pandemic with over 750 million infections and more than 6.9 million deaths, as of June 2023. When symptomatic, coronavirus disease (COVID-19) causes a range of mild to severe clinical manifestations, including fever, cough, sore throat, diarrhea, dyspnea, pneumonia, and occasionally acute respiratory distress syndrome, cytokine storm and multisystem organ failure leading to death[1–4].

Pregnant women, as well as their fetuses, are particularly susceptible to respiratory pathogens, such as influenza viruses and coronaviruses, and are at increased risk of serious complications due to the physiological, mechanical, immunological, and endocrine changes induced by pregnancy[5,6]. Reports initially suggested that pregnant women may be more susceptible to SARS-CoV-2 than matched non-pregnant women, despite the methodological limitations of these early studies[7,8]. Initial publications regarding the impact of SARS-CoV-2 infection on pregnancy outcomes were reassuring, with most studies reporting a risk of severe COVID-19 no greater than that of the general population[9,10]. The majority of hospitalized pregnant women with COVID-19 were in the late second or third trimester and had favorable outcomes[9,10]. In accordance, a recent systematic meta-analysis including 293,152 pregnant and recently pregnant women with COVID-19, compared with 2,903,149 non-pregnant women infected with COVID-19, showed no statistical difference when comparing acute respiratory distress syndrome, major organ failure or all-cause mortality[11]. However, there was a higher rate of intensive care unit admission and need for invasive ventilation or extracorporeal membrane oxygenation in pregnant and recently pregnant women[12]. Similar findings have been described in other international multicentric cohort studies and meta-analyses[12–17]. Higher rates of preterm birth and adverse perinatal outcomes, including stillbirths, were also observed in pregnant women with COVID-19 compared with non-infected pregnant women[11], while vertical transmission of SARS-CoV-2 was found to be a rare event[18]. The rise in preterm births may be partially explained by medical decisions of delivery to treat critically ill mothers, whereas the increase of pre-eclampsia and stillbirth may reflect SARS-CoV-2-induced placental inflammatory lesions[19].

Disease severity in infected pregnant women was associated with risk factors similar to those previously reported for COVID-19, such as high body mass index, preexisting comorbidities, high maternal age and pregnancy-specific conditions such as pre-eclampsia and gestational diabetes[11,20]. Moreover, the increased risk of severe COVID-19 during pregnancy may be due to mechanical changes, in particular decreased lung volume as the fetus grows, changes of the immune system, and increased risk for thromboembolic disease[5,21]. Finally, lower rates of SARS-CoV-2 vaccination in the pregnant population compared with non-pregnant women of reproductive age[22]—despite the lack of specific risk of COVID-19 vaccine during pregnancy[23]—may also explain a worse prognosis.

Therefore, the direct and indirect consequences of SARS-CoV-2 infection are a risk for pregnancy and offspring. How pregnant women and their fetuses respond to SARS-CoV-2 infection is still under investigation, as fetal development may be affected even in the absence of direct fetal infection, and as a fetal response to maternal infection is likely in this circumstance.

We therefore sought to investigate the transcriptome and phenotype changes in fetal cord blood cells induced by maternal SARS-CoV-2 infection, and to study the molecular triggers leading to a potential fetal response. Overall, we observe an erythroid cell signature along with hypoxia pathway activation in cord blood cells from symptomatic infected mothers compared with control pregnant women. This indicates a fetal hematopoietic response to symptomatic COVID-19 in pregnant mothers in the absence of vertically transmitted SARS-CoV-2 infection.

## Methods

**Study population and data collection**. We conducted a retrospective single-center cross-sectional study in compliance with good clinical practice and the Declaration of Helsinki.

Cord blood samples from COVID- and COVID+ pregnant patients were collected at delivery at Cochin maternity department (Paris, France) between April 2020 and April 2021. Inclusion criteria were to be over 18 years old, able to provide consent, and be COVID− or COVID+ (as defined below). The exclusion criteria were: refusal of consent. Patients' medical histories, treatments, laboratory data, and information related to pregnancy were extracted from medical files. Patients did not undergo specific medical examination or tests or follow-up for this study, except for cord blood collection. The STROBE guideline for reporting cross-sectional studies was followed.

Maternal illness severity was defined as follows: (1) negative (COVID-): no COVID-19 symptom at delivery or during pregnancy, negative nasopharyngeal swab PCR test for SARS-CoV-2 infection when performed, (2) asymptomatic COVID-19 (COVID+A): positive nasopharyngeal swab PCR test for SARS-CoV-2 infection but no symptoms of COVID-19 at delivery or during pregnancy; (3) symptomatic COVID-19 (COVID+S): positive nasopharyngeal swab PCR test for SARS-CoV-2 infection with symptoms such as fever, cough, fatigue, and dyspnea at delivery or during pregnancy.

**Cord blood mononuclear cells isolation**. At delivery, 20 mL of cord blood were drawn in ACD-B tubes (Vacuette, Greiner Bio-one) containing citric acid, trisodium citrate, and dextrose. At least 1 mL of undiluted plasma was harvested after centrifugating the tubes at $900 \times g$ for 12 min at room temperature. Then, cord blood was diluted with PBS (Gibco) and carefully layered over 15 mL of Ficoll-Paque Plus density 1.077 g/mL (GE Healthcare) in a 50 mL Blood Sep Filter tube (Dutscher). Tubes were centrifuged at $1000 \times g$ for 10 min at room temperature in a swinging-bucket rotor without brake, according to the manufacturer's instructions. The mononuclear cell layer was transferred to a 50-mL tube and washed twice using RPMI-1640 media (Gibco) containing 10% fetal bovine serum (ThermoFisher Scientific). Cells were divided into three groups at the end of the isolation process: (1) immediate lysis of fresh cells for total RNA extraction; (2) immediate freezing at −80 °C (without freezing media) for total protein extraction; and (3) immediate freezing in freezing media (90% fetal bovine serum + 10% DMSO (Sigma-Aldrich)) for FACS analyses and methylcellulose culture.

**RNA extraction and sequencing**. Fresh cord blood mononuclear cells were lysed and total RNA was extracted using NucleoSpin RNA Mini Kit for RNA purification (Macherey-Nagel), according to the manufacturer's instructions. RNA sequencing was performed and analyzed at Genom'IC Platform of Cochin Institute (Paris, France). Briefly, after RNA extraction, RNA concentrations were obtained using a fluorometric Qubit RNA assay (Life Technologies). The quality of the RNA (RNA integrity number) was determined on an Agilent 2100 Bioanalyzer (Agilent Technologies) according to the manufacturer's instructions. To construct the libraries, 250 ng of high-quality total RNA sample (RIN ≥7.7) were processed using NEBNext Ultra II RNA Library Prep Kit (New England BioLabs) according to the manufacturer's instructions. Briefly, we purified poly-A-containing mRNA molecules, then we fragmented and reverse-transcribed them using random primers. Libraries were quantified by qPCR using the KAPA Library

Quantification Kit for Illumina Libraries (KapaBiosystems), and library profiles were assessed using the DNA High Sensitivity LabChip kit on an Agilent Bioanalyzer. Libraries were sequenced on a Nextseq 500 instrument (Illumina) using 75 base-lengths read V2 chemistry in a paired-end mode. RNA Integrity Numbers and sequencing depth (in millions of reads) for each analyzed sample are presented in Supplementary Table 1.

**Flow cytometry**. Cord blood mononuclear cells were thawed as previously described[24]. Briefly, cryovials were submerged halfway in a 37 °C water bath for 60 s. One mL of warmed RPMI-1640 media containing 10% fetal bovine serum (complete RPMI) was added slowly dropwise to the cells. Cells were then poured into a 15 mL conical tube containing 5 mL of complete RPMI and incubated in a 37 °C water bath for 5 min. After centrifugation at 330 × g at room temperature for 10 min, 1 mL of complete RPMI with 50 U/ml of DNase (Sigma-Aldrich) was added to the cell pellet and cells were incubated at 37 °C in a $CO_2$ incubator for 1 h. Cells were then processed for immunophenotyping.

Cells were stained with antibodies (CD110-PE, CD36-FITC, CD19-ECD, CD123-PC5.5, CD38-PE-Cy7, CD10-APC, CD34-APC-A700, CD71-APC-Vio770, CD45Ra-BV421 and CD45-KrO) at 1:100 dilution (Supplementary Table 2). After 20 min incubation at room temperature, samples were washed once in PBS. Cells were then resuspended in 500 μL of PBS. A Navios flow cytometer (Beckman Coulter) was used to acquire data (10 colors, 3 lasers (5 + 3 + 2 configuration)). The sensitivity of the flow cytometer was controlled daily using Flow-Set Pro beads (Beckman Coulter). Data were analyzed using Kaluza 2.1 version software (Beckman Coulter). All multi-parameter flow cytometry analyses were performed blinded of COVID-19 infection status.

We sorted CD34+ hematopoietic stem and progenitor cells (HSPC) and gated on CD38 expression to identify 2 populations: CD34+ CD38- and CD34+ CD38+ . In CD34+ CD38- population, we identified multipotent progenitor cells (HSC/MPP, CD45Ra-CD10-), lymphoid-primed multipotent progenitor (LMPP, CD45Ra+ CD10-) and multi-lymphoid progenitor (MLP, CD45Ra+ CD10 + ). The HSC/MPP population was then separated into 3 different populations according to CD71 and CD110 expression: MPP F1 (CD71-CD110-), MPP F2 (CD71+ CD110-) and MPP F3 (CD71+ CD110+ ). The CD34+ CD38+ CD10- population was separated into granulocyte–monocyte progenitors (GMP, CD45Ra+ ), common myeloid progenitors (CMP, CD45Ra-CD123+ ) and megakaryocyte-erythrocyte progenitors (MEP, CD45Ra-CD123−). Finally, the MEP population was separated into 3 different populations according to CD71 and CD110 expression: MEP F1 (CD71-CD110−), MEP F2 (CD71+ CD110−), MEP F3 (CD71+ CD110+ ).

**Quantification of hematopoietic progenitors clonogenicity in semi-solid cultures**. Hematopoietic HSPC (CD34+) were isolated from cord blood mononuclear cells by immunomagnetic enrichment using CD34 MicroBead Kit (Miltenyi Biotec).

To perform colony cultures, 1000 cells/ml per condition of CD34+ progenitors were seeded in methylcellulose medium (H4100 Methocult media, StemCell Technologies) supplemented with 12.5% bovine serum albumin (Sigma-Aldrich) (neutralized with 7.5% sodium bicarbonate (Sigma-Aldrich)), 30% fetal bovine serum, 1 mM 2-mercaptoethanol (Sigma-Aldrich), 1% L-glutamine (Gibco), and in the presence of a cocktail of recombinant cytokines containing 1 U/mL EPO, 20 ng/mL TPO, 25 ng/mL SCF, 10 ng/mL IL-3, 10 ng/mL FLT3-L, 20 ng/mL G-CSF, 10 ng/mL IL-6 (all cytokines were purchased from Pepro Tech). Colony-forming units for granulocytes, erythrocytes, monocytes, megakaryocytes (CFU-GEMM), burst-forming units-erythroid and colony-forming units-erythroid (BFU-E/CFU-E), and colony-forming units for granulocytes and macrophages (CFU-GM) were quantified after 14 days of culture.

**Western blot**. Total proteins were extracted from cord blood mononuclear cells using RIPA buffer (Sigma-Aldrich) supplemented with cOmplete protease inhibitors cocktail (Roche) and PhosSTOP phosphatases inhibitors (Roche). Lysates were cleared by centrifugation at 14,000 × g for 10 min at 4 °C. Protein concentrations were determined by the Lowry method, with BSA as standard using Modified Lowry Protein Assay Kit (Pierce) according to the manufacturer's instructions. The primary antibodies used for immunoblotting were as follows: anti-VHL (Cell signaling, 68547S, 1:1000), and anti-ACTIN (Cell Signaling, 12262S, 1:1000). Primary antibody-probed blots were visualized with appropriate horseradish peroxidase-coupled secondary antibody at 1:10,000 dilution (Jackson ImmmunoResearch) using Pierce ECL Western Blotting Substrate (Thermo Scientific) according to the manufacturer's instructions. A ChemiDoc Touch Imaging System (Bio-Rad Laboratories) was used to detect protein bands. Processing of western blot images and quantification of ECL signal were done using ImageJ and normalized to the loading control (ACTIN).

**Measurement of plasmatic cytokines**. ProcartaPlex human 8-plex assays (ThermoFisher Scientific) were used to measure cytokine levels (IFN-α, IL-1α, IL-1β, IL-10, IL-6, IL-8, calprotectin (S100A8/A9), TNF-α) in the plasma of cord blood according to the manufacturer's instructions. They were analyzed with a Bioplex200 device (Bio-Rad Laboratories) and Bio-Plex Manager 6.1 Software (Bio-Rad Laboratories). Each plasma sample was assayed twice with the average value taken as the final result.

Granulocyte-macrophage colony-stimulating factor (GM-CSF) protein concentration was measured in the plasma of cord blood by GM-CSF Human ELISA Kit (Invitrogen) according to the manufacturer's instructions.

**Computational analyses**. Analysis of RNA sequencing was performed using R version 3.6.1 (https://www.r-project.org). Fastq files were aligned using STAR algorithm (version 2.7.6a), on the Ensembl release 101 reference. Reads were count using RSEM (v1.3.1), and statistical analyses on the read counts were performed with R (version 3.6.3) and DESeq2 package (DESeq2_1.26.0) to determine the proportion of differentially expressed genes between two conditions.

We used the standard DESeq2 normalization method (DESeq2's median of ratios with the DESeq function), with a pre-filter of reads and genes (reads uniquely mapped on the genome, or up to 10 different loci with a count adjustment, and genes with at least ten reads in at least three different samples). Following the package recommendations, we used the Wald test with the contrast function and the Benjamini–Hochberg false discovery rate control procedure to identify the differentially expressed genes. Analyses of canonical pathways and upstream regulators were performed using Ingenuity Pathway Analysis (IPA) software (Qiagen). An adjusted P value threshold of 0.05 was applied. No log2FC threshold was applied for these analyses. Identification of putative cell types was performed using Human Gene Atlas (Enrichr)[25].

**Statistics and reproducibility**. Prism 8 software (GraphPad) was used to perform statistical analyses. No statistical method was used to predetermine sample size. The number of biologically independent samples (n, patients) is indicated in the figure legends. At least three different patients were analyzed for each experiment. Shapiro–Wilk test was used to test the normality of

the data. The appropriate statistical method to correct for multiple comparison was used when required, as recommended by Prism software.

**Study approval**. Patients signed a written consent prior to participation. The study complied with all relevant ethical regulations and was approved by the ethics committee of the APHP (ancillary study of COVIPREG study, APHP 200448/N° IDRCB: 2020-A00924-35).

**Reporting summary**. Further information on research design is available in the Nature Portfolio Reporting Summary linked to this article.

## Results

**Clinical characteristics of study participants**. Clinical data of women included in this study, delivery outcomes and fetal characteristics are shown in Table 1. We included 17 pregnant women in our cohort, of which 12 had SARS-CoV-2 infection confirmed by nasopharyngeal swab PCR test. Three of these 12 patients did not present any COVID-19 symptoms at the time of delivery or before (asymptomatic COVID-19, COVID+A), whereas 9 presented mild to severe COVID-19 symptoms such as fever, cough, fatigue or dyspnea (symptomatic COVID-19, COVID+S). Moderate COVID-19 was defined by the presence of dyspnea, and severe COVID-19 by the presence of pneumonia. Three severe cases were admitted to the intensive care unit and required high-flow nasal oxygen or endotracheal intubation. Five pregnant women without any signs of past or current SARS-CoV-2 infection were included as controls (COVID−). There was no significant difference in maternal age at delivery between the three groups, although COVID− women tended to be older (39.8 ± 5.4 years old versus 34.7 ± 2.3 for COVID+A and 32.3 ± 7.4 for COVID+S patients respectively, Table 1). Maternal comorbidities were similar except an increased proportion of overweight patients in COVID+S as previously reported[11,20]. None of the patients had diabetes, arterial hypertension or pre-eclampsia. Four patients had autoimmune conditions (thyroiditis for 3 patients and vasculitis for 1 COVID+S patient) that were successfully treated/or and in stable condition or in remission during pregnancy. Pregnancy terms were similar in all three groups (275.8 ± 5.4 days in COVID− women, 269.3 ± 4.0 for COVID+A women, and 262.8 ± 28.8 for COVID+S women, respectively). The mean delay between SARS-CoV-2 PCR positivity and delivery was of 10.0 ± 9.9 days in COVID+A patients, and of 17.1 ± 18.2 days in COVID+S ones (Table 1).

All women successfully delivered live newborns. Fetal characteristics did not differ significantly between SARS-CoV-2-positive and negative mothers. However, birthweight, Apgar score and cord blood pH tended to be lower in newborns from COVID+S patients as compared with newborns from COVID− and COVID+A patients (Table 1). All neonates from infected mothers were negative for SARS-CoV-2 as demonstrated by negative PCR on throat swab and cord blood, showing the absence of SARS-CoV-2 vertical transmission in our cohort. Lastly, nucleocapsid and spike-specific IgG were detected in the cord blood of 1 COVID+A patient and 4 COVID+S patients (Table 1). Because the inclusion period was from April 2020 to April 2021, none of the women had been vaccinated against SARS-CoV-2 and all samples were collected before the emergence of the Delta and Omicron variants.

**Symptomatic maternal SARS-CoV-2 infection during pregnancy induces a fetal hematopoietic response**. To determine whether SARS-CoV-2 infection in pregnant women triggered a specific fetal response, we isolated cord blood mononuclear cells

and immediately analyzed their transcriptomes by RNA sequencing (Supplementary Table 1). Unsupervised clustering by principal components analysis (PCA) and hierarchical clustering analysis based on Euclidean distance was performed (Fig. 1a and Supplementary Fig. 1a, b). We observed that samples segregated into two distinct groups based on gender (Supplementary Fig. 1a). Using DESeq2 which allows such factor to be modeled during statistical analysis, we identified differences primarily due to COVID-19 by strongly reducing the effect of gender. After gender correction, COVID+S samples were mostly grouped together, while COVID− and COVID+A clustered together, indicating similarity between the latter two groups (Fig. 1a and Supplementary Fig. 1b). Indeed, we compared COVID- and COVID+A transcriptomes and found no significant difference in gene expression (Supplementary Fig. 1c). We therefore decided to combine these 2 groups (COVID−/ + A) and to compare their gene expression profile to COVID+S samples. Volcano plots allowed to identify significant gene expression differences between COVID−/ + A and COVID+S (Fig. 1b). In total, 1035 genes were significantly upregulated and 98 were downregulated in COVID+S cord blood mononuclear cells compared to COVID− and COVID+A ones. Among the highly upregulated genes, we noted several transcripts involved in erythroid differentiation, such as *CD71*, *CD235a*, and *TFRC*. To unveil which cord blood populations were modified during symptomatic COVID-19, we sought to deconvolute their transcriptome. We used a recently published dataset of 21,306 progenitor cells from human cord blood obtained using single-cell RNA sequencing[26]. In this work, Zheng et al. have computationally reconstructed early hematopoietic fate transitions from cord blood and identified 2 intermediate states originating from hematopoietic stem cells (HSC) and multipotent progenitors (MPP): an erythro-myeloid progenitor (EMP) able to differentiate into erythroid and basophil/eosinophil/mast lineages, and a lymphoid-primed multipotent progenitor (LMPP) able to give both lymphoid and selected myeloid populations (neutrophils and monocytes). They found 517 genes consisting in 9 gene modules that correlated with the temporal progression of cell fates[26] (Supplementary Data 2). Using these cell signatures, we found a significant enrichment of EMP and erythroid lineages in COVID+S as compared with COVID− and COVID+A (Fig. 1c). Accordingly, analyses of Human Gene Atlas signatures confirmed a strong upregulation of CD71+ early erythroid signature in COVID+S cord blood cells, as well as of CD105+ endothelial gene signature (Fig. 1d). Ingenuity Pathway Analysis showed an enrichment of genes involved in cell cycle progression of myeloid cells. In particular, important genes for HSC proliferation and self-renewal, such as *TAL1* and *MYBL2*, were identified in COVID+S samples[27,28] (Fig. 1e).

Altogether, these results demonstrate that symptomatic SARS-CoV-2 infection of pregnant women induces profound changes in the gene expression profile of fetal cord blood mononuclear cells related to an expansion of hematopoietic progenitors skewed towards erythroid differentiation.

**Symptomatic maternal COVID-19 leads to an upregulation of multipotent progenitors in fetal cord blood cells and favors progenitors involved in erythropoiesis**. Single-cell analyses have revealed that the transcriptional networks governing commitment during hematopoiesis are relatively conserved between cord blood and bone marrow[26,29]. Therefore, to study the phenotype of cord blood immature hematopoietic cells and to verify if erythroid-committed hematopoietic stem and progenitor cells (HSPC) were increased in COVID+S samples, we used a well-established flow cytometry workflow based on a new hematopoietic hierarchical scheme previously described[30]. This strategy allows the relative quantification of HSPC with high erythroid differentiation

**Table 1 Maternal and fetal clinical characteristics.**

| | SARS-CoV-2 status | | | P value | | |
|---|---|---|---|---|---|---|
| | Negative (COVID−, n = 5) | Positive asymptomatic (COVID+A, n = 3) | Positive symptomatic (COVID+S, n = 9) | COVID− vs COVID+A | COVID− vs COVID+S | COVID+A vs COVID+S |
| *Maternal characteristics* | | | | | | |
| Age, mean ± SD, years | 39.8 ± 5.4 | 34.7 ± 2.3 | 32.3 ± 7.4 | 0.527* | 0.125* | 0.848* |
| Comorbidities, n (%) | | | | | | |
| Pregravid BMI ≥ 25 kg/m$^2$ | 1 (20) | 0 (0) | 4 (44) | >0.999° | 0.580° | 0.491° |
| Diabetes | 0 (0) | 0 (0) | 0 (0) | >0.999° | >0.999° | >0.999° |
| Arterial hypertension | 0 (0) | 0 (0) | 0 (0) | >0.999° | >0.999° | >0.999° |
| Dyslipidemia | 0 (0) | 0 (0) | 1 (11) | >0.999° | >0.999° | >0.999° |
| Tobacco or cannabis use | 0 (0) | 0 (0) | 0 (0) | >0.999° | >0.999° | >0.999° |
| Auto-immune condition | 2 (40) | 0 (0) | 2 (22) | 0.464° | 0.580° | >0.999° |
| Pre-eclampsia/gestational hypertension | 0 (0) | 0 (0) | 0 (0) | >0.999° | >0.999° | >0.999° |
| Parity, n (%) | | | | | | |
| Primiparous | 2 (40) | 2 (67) | 3 (33) | >0.999° | >0.999° | 0.523° |
| Multiparous | 3 (60) | 1 (33) | 6 (67) | | | |
| Term, mean ± SD, days | 275.8 ± 5.4 | 269.3 ± 4.0 | 262.8 ± 28.8 | 0.915* | 0.554* | 0.897* |
| Duration of COVID-19 infection at delivery, mean ± SD, days | NA | 10.0 ± 9.9 | 17.1 ± 18.2 | - | - | 0.764` |
| *Fetal characteristics* | | | | | | |
| Newborn gender | | | | | | |
| Male | 2 (40) | 2 (67) | 4 (44) | >0.999° | >0.999° | >0.999° |
| Female | 3 (60) | 1 (33) | 5 (56) | | | |
| Birthweight, mean ± SD, g | 3142.0 ± 355.9 | 3253.0 ± 592.8 | 2826.0 ± 592.4 | 0.956* | 0.553* | 0.474* |
| Neonatal death, n (%) | 0 (0) | 0 (0) | 0 (0) | >0.999° | >0.999° | >0.999° |
| Apgar score, mean ± SD | 10.0 ± 0.0 | 10.0 ± 0.0 | 8.7 ± 1.7 | >0.999* | 0.197* | 0.309* |
| Cord blood pH, mean ± SD | 7.30 ± 0.07 | 7.34 ± 0.10 | 7.21 ± 0.10 | 0.822* | 0.315* | 0.141* |
| SARS-CoV-2 PCR positivity (throat swab), n (%) | NA | 0 (0) | 0 (0) | - | - | - |
| SARS-CoV-2 PCR positivity (cord blood), n (%) | 0 (0) | 0 (0) | 0 (0) | - | - | - |
| SARS-CoV-2 IgG serology positivity (cord blood), n (%) | 0 (0) | 1 (33) | 4 (44) | - | - | - |

Data information: Data are presented as the number of cases with percentage of total events or means ± SD.
Statistical analyses were performed with ordinary one-way ANOVA with Tukey's multiple comparisons test (*), Fisher's exact test (°), or Mann–Whitney test (`).

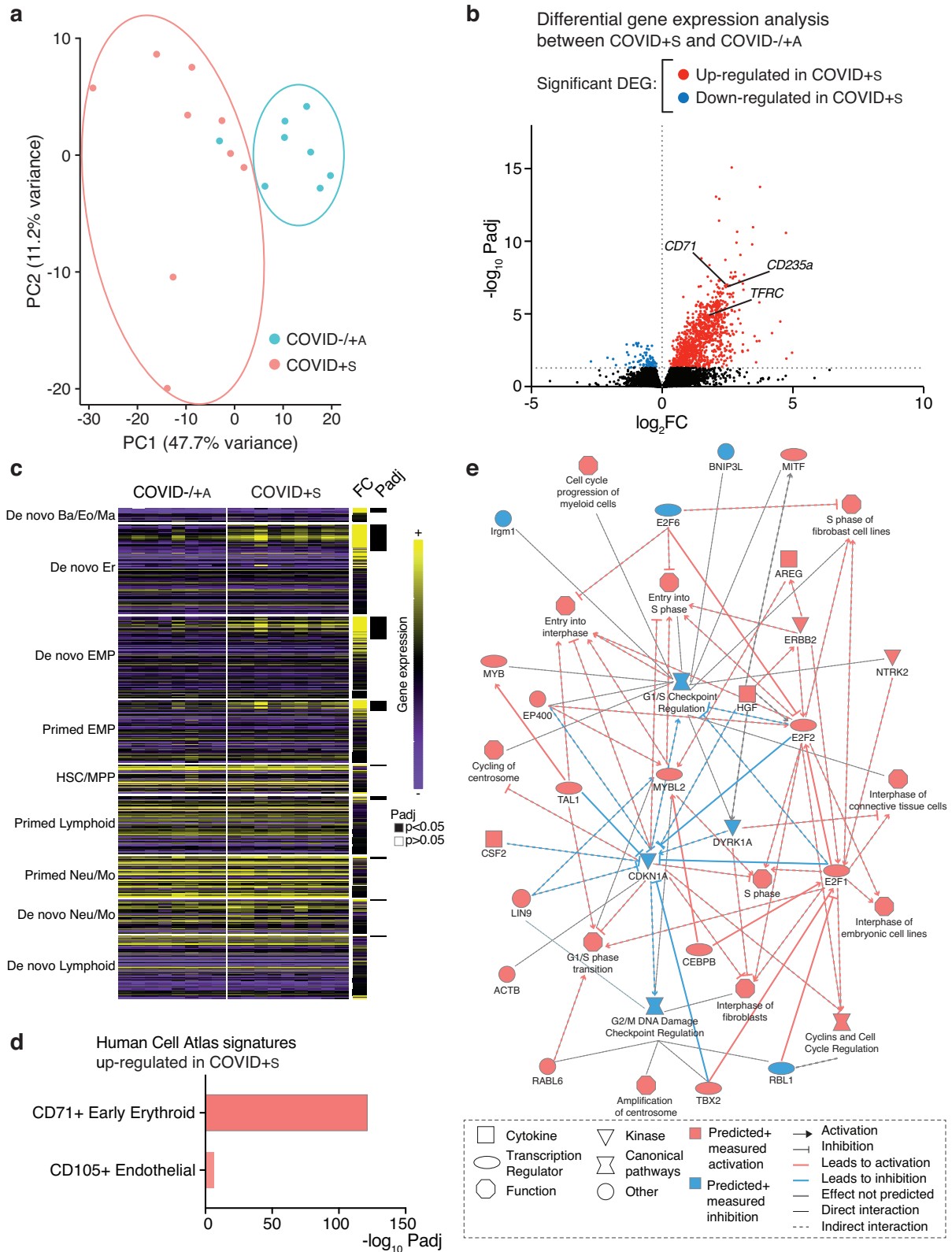

potential among total CD34+ cells (Fig. 2a). We labeled cord blood mononuclear cells of 3 COVID− samples and 3 severe COVID+S samples with a panel of 10 antibodies (Supplementary Fig. 2a). After gating CD34+ cells, we identified the different HSPC subpopulations among both CD38- and CD38+ fractions of cells (Fig. 2a and Supplementary Fig. 2a). The percentage of MPP F1, which are known to have a multipotent differentiation

potential, was 2.5 times higher in cord blood from COVID+S patients than in COVID− patients (respectively 18.93 ± 2.88 % versus 7.55 ± 4.32%, $P = 0.019$) (Fig. 2b). We also observed an increased amount of the specific HSPC subpopulations with high erythroid differentiation potential such as MPP F2 and MPP F3 progenitors, as well as of megakaryocyte-erythrocyte progenitors MEP F1, MEP F2 and MEP F3 in cord blood from severe COVID

**Fig. 1 Cord blood mononuclear cells from symptomatic COVID-19 mothers are enriched for erythroid progenitors. a** Principal Component Analysis (PCA) plot of all samples based on the 500 most expressed genes. Blue dots correspond to COVID− and COVID+A samples (COVID−/ + A), and red dots correspond to COVID+S samples. **b** Volcano plot showing significantly upregulated (in red) and downregulated (in blue) differentially expressed genes (DEG) in COVID+S samples compared with COVID−/ + A samples. **c** Heatmap displaying the expression of the defining genes of the 9 modules displayed in[26] (Supplementary Data 2) between COVID+S and COVID−/ + A samples. Mean fold change (FC) in COVID+S versus COVID−/ + A samples, and adjusted P values (Padj) are shown. **d** Human Cell Atlas signatures enrichment analysis of significantly upregulated genes in COVID+S samples compared with COVID−/ + A samples. **e** Graphical summary obtained upon transcriptional analysis of significantly upregulated genes in COVID +S samples compared with COVID−/ + A samples using Ingenuity Pathway Analysis software. Data information: **a–e** RNA-seq was performed in cord blood mononuclear cells from COVID−/ + A mothers (n = 8) or from COVID+S mothers (n = 9) harvested at delivery. An adjusted p value below 0.05 was considered statistically significant.

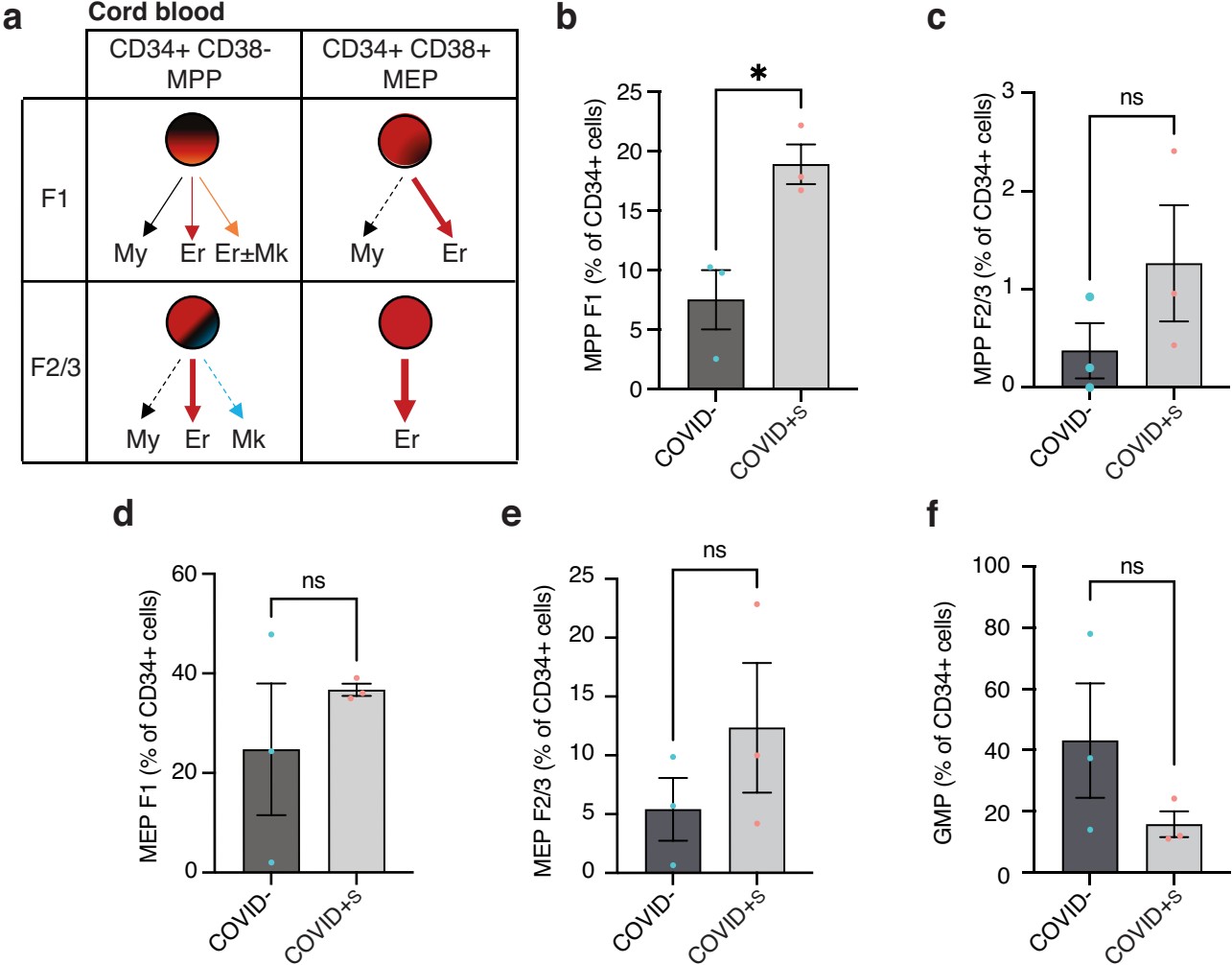

**Fig. 2 Flow cytometry analysis of cord blood mononuclear cells reveals an enrichment of erythroid-committed hematopoietic progenitors upon symptomatic COVID-19 in mothers. a** Different fates (myeloid (My), erythroid (Er) and/or megakaryoid (Mk)) of CD34+ CD38- MPP F1 and F2/3 and CD34+ CD38+ MEP F1 and F2/3 according to ref. [30]. **b–f** Quantification of MPP F1 (**b**), MPP F2 and F3 (F2/3) (**c**), MEP F1 (**d**), MEP F2/3 (**e**), and GMP (**f**) subpopulations as percentages of CD34+ cells in COVID− (n = 3 patients) and severe COVID+S (n = 3 patients) cord blood mononuclear cells. Data are presented as means with SEM and individual values. Statistical analyses were performed with two-tailed unpaired t test (**b**, **c**, **e**, **f**) or two-tailed unpaired t test with Welch's correction (**d**). ns not significant; *P value < 0.05. Source data are available in Supplementary Data 1.

+S patients, yet these results were not statistically significant (Fig. 2c–e). In contrast, granulocyte–monocyte progenitors (GMP) tended to be less abundant in COVID+S patients than in COVID− patients (Fig. 2f). No significant changes were observed in the proportion of common myeloid progenitors (CMP), LMPP, multi-lymphoid progenitors (MLP) or B/NK progenitors (BNKPro) (Supplementary Fig. 2b–e).

This broad phenotypic characterization of hematopoietic cord blood progenitors suggests that HSPC with multipotent and high erythroid differentiation potential are increased in the cord blood of pregnant patients with severe COVID-19 compared to non-infected patients.

**Symptomatic maternal SARS-CoV-2 infection is associated with an increased clonogenicity of erythroid progenitors.** Next, we investigated the clonogenicity of fetal cord blood progenitors of COVID− and severe COVID+S women by seeding at low-

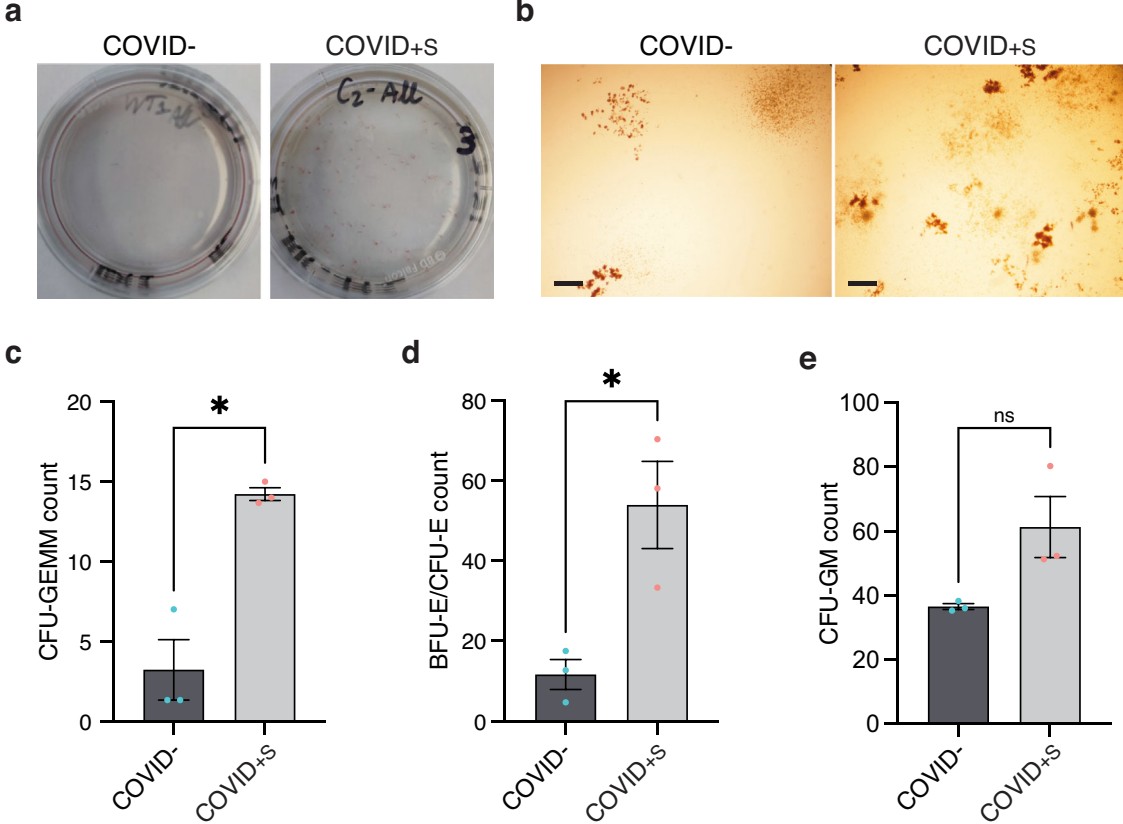

**Fig. 3 Symptomatic maternal COVID-19 infection stimulates clonogenic growth of cord blood hematopoietic progenitors. a**, **b** Representative images of colonies grown for 14 days in methylcellulose from CD34+ HSPC sorted from COVID− and COVID+S samples. Photographs (**a**) and pictures taken at ×4 magnification (**b**) of representative 35-mm dishes are shown. Scale bar: 500 μm. **c–e** Quantification of colony-forming units for granulocytes, erythrocytes, monocytes, megakaryocytes (CFU-GEMM) (**c**), burst-forming units-erythroid and colony-forming units-erythroid (BFU-E/CFU-E) (**d**), and of colony-forming units for granulocytes and macrophages (CFU-GM) (**e**) in COVID− (*n* = 3 patients) and COVID+S (*n* = 3 patients) samples. Data are presented as means with SEM and individual values. Statistical analyses were performed with two-tailed unpaired *t* test (**d**) or two-tailed unpaired *t* test with Welch's correction (**c**, **e**). ns not significant; *P value < 0.05. Source data are available in Supplementary Data 1.

density sorted CD34+ HSPC in methylcellulose medium supplemented with cytokines and growth factors (EPO, TPO, SCF, IL-3, FLT3-L, G-CSF, and IL-6). After 14 days of culture, colonies were present more frequently in COVID+S as compared with COVID− samples (Fig. 3a, b). The number of colony-forming units for granulocytes, erythrocytes, monocytes, megakaryocyte (CFU-GEMM) and burst-forming units-erythroid and colony-forming units-erythroid (BFU-E/CFU-E) was significantly higher in COVID+S cord bloods (CFU-GEMM: $3.2 \pm 3.3$ in COVID− versus $14.2 \pm 0.7$ in COVID+S, $P = 0.024$; BFU-E/CFU-E: $11.6 \pm 6.5$ in COVID− versus $53.9 \pm 18.8$ in COVID+S, $P = 0.022$) (Fig. 3c, d). There was an increase of colony-forming units for granulocytes and macrophages (CFU-GM) in COVID+S samples, albeit not significant ($36.6 \pm 1.6$ in COVID− versus $61.3 \pm 16.5$ in COVID+S, $P = 0.120$) (Fig. 3e).

These results showed an increased clonogenic potential of multipotent and erythroid cord blood hematopoietic progenitors upon maternal severe COVID-19.

**Hypoxia is likely to trigger an erythropoiesis response of fetal cord blood cells.** Then, we sought to understand what could trigger the increased number and clonogenicity of hematopoietic progenitors, especially of erythroid-biased ones in the cord blood of SARS-CoV-2-infected symptomatic mothers.

First, we measured plasma levels of 8 cytokines (IFN-α, IL-1α, IL-1β, IL-10, IL-6, IL-8, S100A8/A9 (calprotectin), and TNF-α),

which were shown to be upregulated in the blood of severe COVID-19 patients[31–33]. None of these circulating proteins significantly differed between COVID+S and COVID− or COVID+A samples (Fig. 4a–h). We also measured the level of granulocyte-macrophage colony-stimulating factor (GM-CSF) and found no difference between COVID+S and COVID− or COVID+A samples (Fig. 4i). Therefore, these results do not support the hypothesis of stress hematopoiesis in response to inflammatory cytokines associated with SARS-CoV-2 infection.

A longitudinal multi-omics study of COVID-19 patients at different disease stages was performed by Bernardes et al.[34]. Using bio-informatic tools, they found 10 modules (M1 to M10) of co-expressed genes following distinct expression patterns throughout COVID-19 infection and recovery (Supplementary Data 3). They identified in critically ill patients a significant increase of erythropoiesis with features of hypoxic signaling, corresponding to module M7[34]. We used these modules to explore the transcriptome of cord blood mononuclear cells from our patients, and found a significant enrichment of module M7 in COVID+S as compared with COVID− and COVID+A (Supplementary Fig. 3), supporting the hypothesis of an erythroid response in the cord blood of symptomatic mothers infected with SARS-CoV-2. Of note, the outcome was favorable in all the COVID+S mothers, even the 3 severe ones, suggesting that this upregulation of module M7 genes in the cord blood cells was not predictive of a fatal disease outcome in the mothers. Then, we evaluated maternal hemoglobin (Hb) content and oxygen saturation (SpO2) at the time of delivery

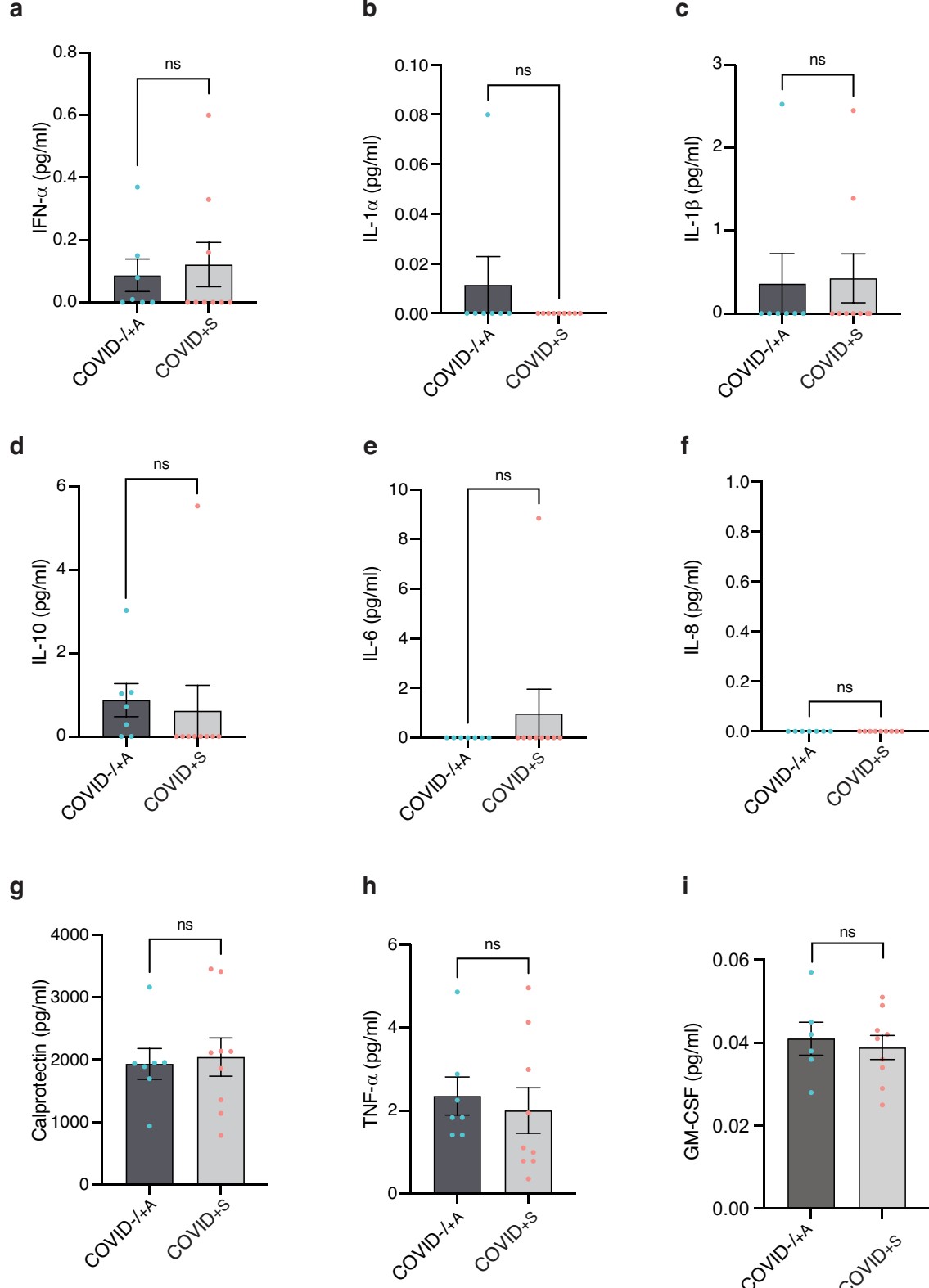

**Fig. 4 Symptomatic maternal COVID-19 infection is not associated with cord blood inflammation. a–i** Quantification of IFNα (**a**), IL-1α (**b**), IL-1β (**c**), IL-10 (**d**), IL-6 (**e**), IL-8 (**f**), calprotectin (S100A8/A9) (**g**), TNF-α (**h**), and GM-CSF (**i**) levels in COVID− / + A (n = 7 patients) and COVID+S (n = 9 patients) samples. Data are presented as means with SEM and individual values. Statistical analyses were performed with Mann–Whitney test with Holm–Sidak's correction. ns not significant. Source data are available in Supplementary Data 1.

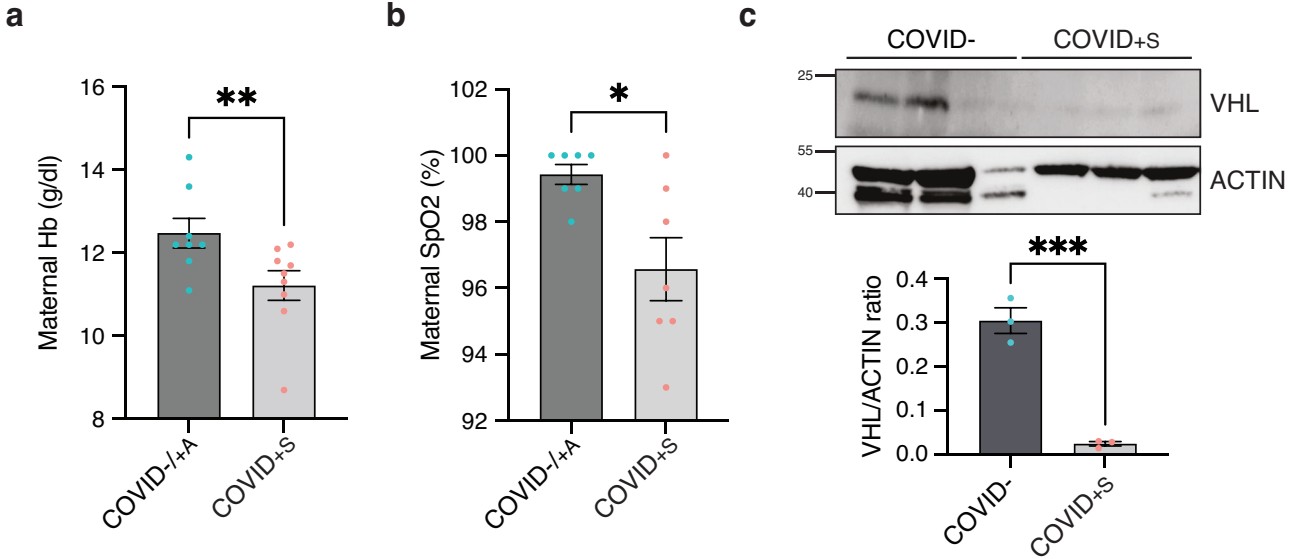

**Fig. 5 Maternal COVID-19-related hypoxia is likely to induce HIF-1α pathway activation in cord blood mononuclear cells. a, b** Maternal hemoglobin level (**a**) and oxygen saturation (**b**) were measured at delivery in COVID−/ + A ($n = 8$ patients) and COVID+S ($n = 9$ patients) women. **c** Representative western blot, with quantification, of VHL in COVID− ($n = 3$ patients) and COVID+S ($n = 3$ patients) samples. Relative protein amounts were normalized to ACTIN levels. Data information: Data are presented as means with SEM and individual values. Statistical analyses were performed with Mann–Whitney test (**a, b**), or two-tailed unpaired *t* test (**c**). *$P$ value < 0.05; **$P$ value < 0.005; ***$P$ value < 0.0005. Uncropped versions of the western blots presented in (**c**) are shown in Supplementary Fig. 4. Source data are available in Supplementary Data 1.

in the mothers of our cohort. As expected, both maternal Hb levels and SpO2 were significantly reduced in COVID+S mothers compared to COVID− or COVID+A mothers (maternal Hb: 12.5 ± 1.0 g/dl in COVID−/ + A versus 11.2 ± 1.1 g/dl in COVID +S, $P = 0.009$; maternal SpO2: 99.4 ± 0.8% in COVID−/ + A versus 96.6 ± 2.5% in COVID+S, $P = 0.033$) (Fig. 5a, b). Thus, we hypothesized that hypoxia may promote a rescue hematopoiesis upon maternal COVID-19 infection. It was not possible to accurately assess the expression of hypoxia-inducible factor-1α (HIF-1α) due to the lability of this protein in normoxia and the specific sampling procedure required for its detection[35]. We therefore evaluated the expression of Von Hippel–Lindau (VHL) protein by western blot. VHL protein targets HIF-1α for proteasome degradation under normoxic conditions, and is the master regulator of HIF activity[36]. VHL was significantly decreased in COVID+S samples (Fig. 5c), supporting the hypothesis that maternal hypoxia is likely to be sensed by the fetus and to trigger fetal stress hematopoiesis.

## Discussion

Our study shows that cord blood cells of symptomatic SARS-CoV-2-infected mothers display at delivery a transcriptional erythroid cell signature as compared with asymptomatic and non-infected mothers. Using flow cytometry and cultures of hematopoietic progenitors, we observed that there was an expansion of different myeloid progenitors, skewed towards the erythroid compartment. There was no evidence of inflammatory cytokines passing to the fetus. Interestingly, we observed features indicating an activation of the hypoxia pathway in these cord blood specimens suggesting that maternal hypoxia may be triggering the fetal stress hematopoiesis we observed. Altogether, these results support the hypothesis of a fetal response to symptomatic COVID-19 in pregnant mothers in the absence of vertically transmitted SARS-CoV-2 infection (Fig. 6).

We observed that transcriptomes obtained from male and female fetuses segregated into two distinct sex-dependent clusters independently from SARS-CoV-2 status. Indeed, it has been

described that the gene expression profile of cord blood cells of different genders cluster separately[37]. Interestingly, this difference has also been shown in stemness-related genes[38]. For transcriptomic analyses, we therefore applied a correction based on sex to overcome this bias and study only the effect of maternal COVID-19.

Severe SARS-CoV-2 infection has been found to induce a dysregulated hematopoiesis in infected adults, characterized by an increased proportion of immature and dysfunctional myeloid cells, especially megakaryocyte progenitors, monocytes and granulocytes, and a reduced number of lymphoid progenitors[31,32,34,39–41]. In accordance, a reduction of host HSC/MPP progenitors was observed, as well as a priming of uncommitted CD34+ HSC/MPP progenitors towards myelopoiesis, in particular megakaryopoiesis and erythropoiesis[34,39,40]. These alterations in hematopoiesis have been linked to more severe outcomes in infected patients[34].

The expression of ACE2—the cell membrane protein targeted by SARS-CoV-2 Spike protein enabling cell entry—, as well as of other host proteins implicated in viral entry, such as CD147, CD26, and TMPRSS2, has been described at the surface of HSC and erythroid precursors[42]. In particular, CD34-CD117+ CD71+ CD235a- early erythroid progenitors are a preferential target of SARS-CoV-2 infection leading to an increased clonogenicity of infected cells and stress erythropoiesis[43]. Another group has confirmed that CD71+ erythroid cells (CEC) are a target of SARS-CoV-2 and that their abundance correlates with COVID-19 severity[44]. Their abundance in the blood of COVID-19 patients also varies with the strain of SARS-CoV-2, as they are more frequently found in patients infected with the Wuhan original strain, than with the Delta or Omicron isolates[45]. These CEC cells are of particular interest as they represent an heterogeneous population of CD71+ CD235a+ erythroid progenitors/precursors with a wide range of immunosuppressive and/or immunomodulatory properties[46,47]. While this is useful to promote feto-maternal tolerance[48], it also leads to a limitation of innate and adaptive immune anti-infectious responses[46,49]. CEC account for ~50–70% of cord blood mononuclear cells[46,50], but

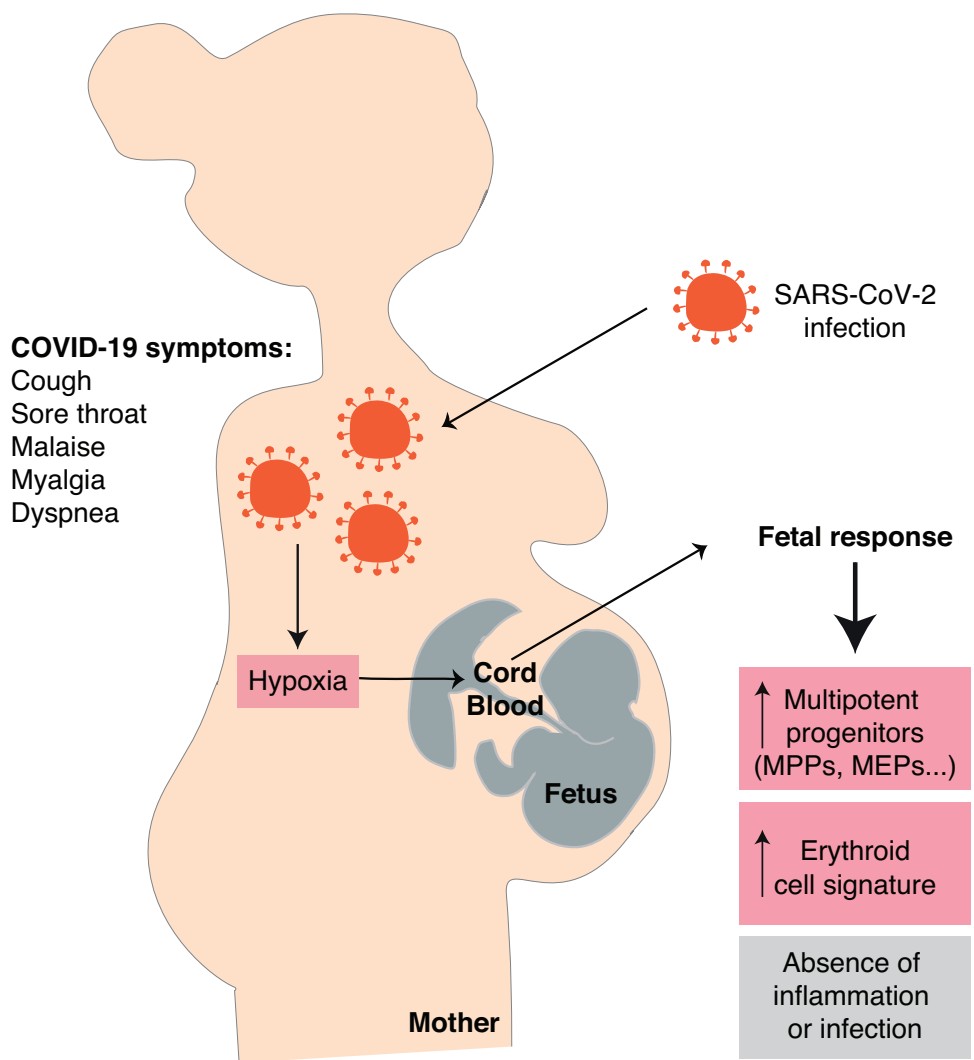

**Fig. 6 In symptomatic SARS-CoV-2-infected mothers, hypoxia is likely to induce a fetal response leading to an expansion of erythroid-committed hematopoietic progenitors.** In pregnant women with symptomatic COVID-19 and respiratory symptoms such as cough and dyspnea, maternal hypoxia is likely to be detected by the fetus and result in a fetal response. An expansion of fetal hematopoietic multipotent progenitors skewed towards erythroid differentiation and a transcriptional erythroid cell signature are observed in umbilical cord blood, while there is no difference in inflammatory cytokines levels, nor evidence of SARS-CoV-2 infection in the offspring.

they are extremely sensitive to hypotonic lysis[51]. Therefore, we were not able to study CEC cells in our samples as we performed FACS analyses and clonogenic cultures after a freeze-and-thaw cycle for practical reasons. Instead, we chose to focus on the cord blood CD34+ compartment as these cells are not sensitive to freezing[52]. Therefore, the CD34+ CD71+ population we measured did not correspond to CEC, but rather to hematopoietic stem cells/progenitors committing to erythroid differentiation.

ACE2 and TMPRSS2 are also expressed at the surface of a population characterized as CD45-CD34+ CD133+ small embryonic-like stem cells of human cord blood able to provide HSC and endothelial progenitor cells. SARS-CoV-2 infection of these cells was reported to activate Nlrp3 inflammasome pathway and pyroptosis[53]. We did not see any expression of SARS-CoV-2 entry receptors ACE2 or transmembrane protease TMPRSS2 in cord blood cells of both non-infected or infected mothers. In agreement, SARS-CoV-2 PCR were negative in cord blood samples from infected mothers. Therefore, it suggests that cord blood mononuclear cells may not be an important site of COVID-19 infection. Of note, several receptors and co-receptors of the SARS-CoV-2 are expressed in human placenta by a

subpopulation of syncytiotrophoblasts in the first trimester and extravillous trophoblasts in the second trimester[54]. SARS-CoV-2 has been detected in around 50% of placentas of infected mothers[55,56], although contradictory results were published[57]. Nonetheless, even in the case of infected placentas, cases of vertical transmission remained rare, suggesting that the placenta is an effective maternal-neonatal barrier against SARS-CoV-2[55].

While cord blood cells were not infected by SARS-CoV-2, we observed that in response to symptomatic maternal SARS-CoV-2 infection, a fetal emergency myelopoiesis and erythropoiesis is triggered, characterized by the presence of more HSC/MPP progenitors in the cord blood. These CD34+ progenitors were able to form significantly more CFU-GEMM and BFU-E/CFU-E colonies in clonogenic assays. Of note, an erythropoiesis signature was also observed at the transcriptomic level in cord bloods from symptomatic but non-critically ill COVID+ mothers. In addition, all women in this cohort had a rapid favorable outcome after delivery. Another study found an increased proportion of differentiation towards BFU-E in cord bloods at term obtained from women previously infected and who recovered from SARS-CoV-2[58]. This could indicate that the erythropoiesis

induction we observed shortly after the infection may be sustained throughout the pregnancy. However, these authors also found less granulocyte-erythrocyte-monocyte-megakaryocyte (CFU-GEMM) and granulocyte-macrophage (CFU-GM) colonies in previously infected women[58]. There was also a reduction of cord blood CD34+ progenitors in women at term who recovered from the infection, that the authors linked to an increased susceptibility to IFN-γ-mediated apoptosis in these cells[58]. Pregnancy induces specific changes in maternal immunological status to enable fetal tolerance through attenuation of T-cell and humoral responses, while maintaining effective responses to pathogens through enhancement of innate responses[59]. Single-cell RNA sequencing of peripheral blood mononuclear cells isolated from recovered pregnant and non-pregnant COVID-19 patients versus matched healthy controls revealed that infected pregnant women display a marked lymphopenia but keep a functional lymphoid immune response. In parallel, they harbor an increased activation and chemotaxis of NK, NKT, and MAIT cells which may be beneficial in the context of the infection[60]. These immune modifications are related to the intensity of COVID-19, as peripheral blood from asymptomatic or paucisymptomatic showed fewer alterations[61]. We did not observe in the cord blood of infected mothers an increase of several cytokines that were previously shown to be upregulated in the blood of severe COVID-19 patients[31–33]. These results are in agreement with another study which only found a moderate elevation in IL-8 concentration in the cord blood of infected mothers[57]. In addition, while T-cell subsets were perturbed in the mother peripheral blood, no change was observed in these immune populations in the cord blood[57]. Another study confirmed the absence of difference in the immunological profile of NK, T, and B lymphocytes and cytokines in cord bloods from pregnant women during the late recovery stage from COVID-19[62]. However, an activation of interferon signaling, plasmacytoid dendritic cells, and NK cells was also observed in cord blood mononuclear cells from infants born to SARS-CoV-2-infected mothers with mild COVID-19 compared with healthy mothers[63,64]. In addition, transcriptional antimicrobial immune responses have been demonstrated in the transcriptome of umbilical cord blood cells in the context of maternal SARS-CoV-2 infection[57]. In conclusion, while SARS-CoV-2 induces prolonged immune modifications in maternal blood, cord blood seems to display only limited variations of cytokines and immune cells.

We observed features showing an activation of a hypoxia-responsive pathway in cord blood cells. Indeed, VHL protein was strongly reduced in COVID+S samples indicating HIF-1α degradation[36]. We propose it could be a causing mechanism of the hematopoietic fetal response. Bone marrow and cord blood are physiologically hypoxic organs, therefore oxygen sensing signaling pathways, such as HIF-1α pathway, play critical roles to regulate hematopoiesis[65,66]. Cord blood HSC/progenitors grown under hypoxic conditions show increased expression of HIF-1α and increased expansion capabilities in vitro and in vivo[67,68], through VEGFR1 stimulation[69,70] and participation of NOTCH-mediated signaling[71]. Hypoxia also leads to a metabolic shift in CD34+ hematopoietic progenitors from umbilical cord blood favoring the development of megakaryocyte-erythrocyte progenitors and erythroid differentiation[72]. In the context of COVID-19, mild hypoxic features, such as an increase in syncytial knots, were reported in the placentas of women in the late recovery stage from COVID-19[62]. Pathological signs indicating poor maternal or fetal vascular perfusion of the placenta were also observed in a significant number of patients, without any impact on the fetal outcome[73,74]. Thus, we hypothesize that maternal hypoxia upon SARS-CoV-2 infection is likely to trigger a fetal stress hematopoiesis. This fetal response could be seen as an adaptive mechanism of the fetus to be more

resilient and adapted to an hypoxic environment, thus limiting the odds of an adverse outcome.

Numerous studies have shown, both in rodents and primates, that during pregnancy there is a mutually beneficial constant bidirectional transfer of cells between fetuses and mothers[75]. In the fetus, transferred maternal cells may increase maternofetal tolerance, be a source of mature immune cells, and help shape the immune system during development[75]. In the mother, transferred fetal cells persist in the maternal bone marrow at low levels for decades after delivery, are well tolerated by the maternal immune system and contain different types of progenitors including hematopoietic and mesenchymal stem cells able to migrate and repair different damaged maternal tissues[75–79]. The transfer of chimeric cells between the mothers and their offspring can be influenced by external factors, including maternal stress and infection. As an example, pertussis toxin-induced inflammation in mothers during pregnancy causes increased cellular trafficking to the fetus[80]. Thus, we could hypothesize that fetal stress hematopoiesis triggered by maternal hypoxia could lead to the release of erythroid progenitors in the maternal circulation to help mothers cope with the infection. While unbiased epidemiological data are difficult to obtain and proper comparisons are missing, COVID-19-related mortality in pregnant women seemed to be lower than those related to sister coronavirus infections such as SARS and MERS[81]. The possibility that fetal progenitors may be transferred to mothers to mitigate the burden of COVID-19 is an exciting possibility that may be difficult to confirm, but may explain why the SARS-CoV-2 outbreak was less severe than expected in pregnant women.

**Limitations of the study**. Our study has several limitations that need to be taken into account. Firstly, we acknowledge the limited sample size and the monocentric recruitment of our cohort. This may lead to potential biases and may not reflect the diversity of the pregnant woman population infected with SARS-CoV-2. In addition, we collected cord blood samples immediately after delivery and processed them immediately to ensure the quality of our samples (the maximum time between delivery and collection was one hour). Consequently, we had to freeze the cells for downstream analyses, such as FACS and methylcellulose cultures, as it would not have been possible to analyze them immediately. As a result, we were unable to quantify and analyze by flow cytometry-specific populations such as CD71+ erythroid cells due to their vulnerability to freezing. Lastly, the difficulty of harvesting more samples, including maternal cells and plasma, has limited our ability to provide more mechanistic experiments to fully support our conclusions.

## Data availability

All data that support the conclusions are available from the corresponding author upon reasonable request. The RNA-seq dataset produced in this study is available at the NCBI Gene Expression Omnibus (GEO) (accession no. GSE224063). Publicly available datasets used in this study were: GEO GSE97104[26] and GEO GSE16177[34]. All the source data underlying the graphs presented in Figs. 2b–f, 3c–e, 4a–i, 5a–c, and Supplementary Fig. 2b–e are available in Supplementary Data 1.

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

## Acknowledgements

We thank all members of the Port Royal Obstetrics Department of Cochin Hospital for their help in collecting human samples. We are grateful to all members of the Cutaneous Biology laboratory for helpful discussions. We thank Sandrine Luce for technical help. We are grateful to the Genom'IC Facility of Institut Cochin, particularly Benjamin Saint-Pierre and Lucie Adoux for their help with RNA-seq analyses. Luminex assay was performed at CYBIO facility of the Cochin Institute with the help of Céline Bertholle. This work was funded by grants to S.A. from the Agence Nationale de la Recherche (ANR19-CE17-0025-04) and to B.O. from INSERM-Fondation Bettencourt Schueller (R20011KS).

## Author contributions

M.A., V.T., S.A., and B.O. conceived the study and designed the experiments with help from V.C., and M.F. M.A., Q.T.N., N.C., R.H.F., M.E.K., M.T., N.H., A.J., and B.O. performed experiments and analyzed data. M.A., R.H.F., S.A., and B.O. wrote the manuscript with inputs from all the authors.

## Competing interests

The authors declare no competing interests.
