## [Peer Review File · Communications Medicine]

Reviewers' comments:

Reviewer #1 (Remarks to the Author):

In this paper Alkobtawi et al. report on their study on fetal response to maternal SARS-CoV-2 infection during pregnancy. The authors investigated the transcriptome and phenotype of cord blood cells in a retrospective single-center cohort study and report an erythroid cell transcriptional signature in cord blood from mothers with symptomatic SARS-CoV-2 infection, as well as an increased erythroid differentiation of fetal hematopoietic multipotent progenitors in quality and quantity.

The authors interpret these findings as possible fetal stress erythropoiesis induced by maternal hypoxia. The study is novel and relevant to the field, as the conclusions would add to existing reports of potential placental problems in SARS-CoV-2 infection.

Overall, the reported claims can be made from the existing transcriptomics, clonogenicity and FACS analysis, and especially the latter is comprehensively performed and described.

However, there are some issues:

1. The title "Enhanced fetal erythropoiesis in response to symptomatic COVID-19 infection during pregnancy" is not differentiating between disease and virus. COVID-19 is the disease, SARS-CoV-2 the virus. There is no COVID-19 infection. Therefore, the title should be corrected, e.g., to „Enhanced fetal erythropoiesis in response to COVID-19 during pregnancy“ or „Enhanced fetal erythropoiesis in response to symptomatic SARS-CoV-2 infection during pregnancy“
2. Because no reviewer token is provided, the generated RNA-seq data (GSE224063) cannot be evaluated.
3. According to the author's description, the cord blood was already centrifuged at 900 g for 12 min before Ficoll-Paque gradient centrifugation. Is this a misunderstanding? Plasma could have been easily collected after Ficoll-Paque PLUS gradient centrifugation.
4. Usually gradient centrifugation by Ficoll-Paque is done for more than 10 min (30-40 min). Was there a specific reason for using only 10 min. A reference for a 10 min protocol is recommended. Was 10 min always employed? Blood from infants after birth may contain early erythroid cells (CD71+, who show a high TAL1 expression, Correia et al. Leukemia 30, 1968–1978 (2016)), that may contaminate the upper buffy coat layer (<https://www.urmc.rochester.edu/medialibraries/urmcmedia/rochester-human-immunology-center/documents/HIC-1-0020Approved.pdf>). Therefore, the question of a possible contamination and pre-analytic error because of rather short centrifugation time would have to be asked. However, if the isolation of mononuclear cells from cord blood was always done in the same way, this would be a balanced error.
5. RNA extraction and sequencing should include some more information, e.g., was the total RNA depleted for ribosomal RNA (likely?). Otherwise, there may not have been the sequencing depth to detect more differences between the groups.
6. Also, the protocol/kit used for RNA-seq library preparation should be named.
7. Why was an outdated version of STAR (2.7.6a) employed (current as of 2022: 2.7.10b, <https://github.com/alexdobin/STAR/blob/master/CHANGES.md>)
8. Why was an outdated version of R (3.6.1) employed (current as of 2022: 4.2.2. <https://stat.ethz.ch/pipermail/r-announce/2022/000680.html>)
9. Which log2FC cut-off was employed before doing downstream analysis (esp. Ingenuity Pathway Analysis)?

10. The authors have made many sophisticated comparisons with existing transcriptome data. For ease of reproduction, the relevant module gene sets should be presented in a supplementary table.

11. The finding that transcriptomes from male and female cord bloods cluster separately (Fig. 1a) is not discussed. While adjustment for gender in SARS-CoV-2 infection is justified a short discussion, taking into account the existing literature is recommended.

Reviewer #2 (Remarks to the Author):

The manuscript by Alkobtawi et al. claims that SARS-CoV-2 infection in pregnant women results in enhanced erythropoiesis. This is in line with previous reports showing the expansion of erythroid precursors/progenitors in the peripheral blood of COVID-19 patients. The descriptive nature of the study is the major limitation. Small sample size and limited mechanistic studies to support their findings.

It is well documented that cord blood is enriched with erythroid precursors/progenitors defined as CD71+ erythroid cells (CECs) (e.g. PMID: 30272151, PMID: 29735482, and PMID: 24196717). Under physiological conditions, the buffy coat (PBMCs) from the cord blood contains 50-70% CECs. So, it would have been highly informative to provide flow plots of the frequency of CECs in different groups.

There is a major concern in the methodology. CECs from cord blood or any other sources do not resist the freeze-and-thaw process. Basically, they get lysed in the freezing media unless they have a hybrid erythroid/myeloid phenotype.

Based on the described methodology that the authors have stored the cells at -80 until use. The integrity of total RNA from CECs is questionable. Because the cells has to be washed off the freezing media prior to RNA isolation. Hence, all the content of CECs will be washed away.

Since erythroid precursors have a very low RNA content, I speculate that samples from COVID-19+ might have had a greater number of erythroid progenitors or nucleated CECs. I think it's impossible to compare CD45+ among the erythroid signature in the bulk seq to address this question. It will be feasible in ScRNASeq.

Regardless of the quality of RNA, there is a possibility that COVID-19 modulates hematopoiesis rather than erythropoiesis. The text/title should be modified accordingly. By any chance did the author compare CD45+ subpopulations among CECs between COVID-19+/- groups by flow cytometry?

I did not see the RIN score for the analyzed samples.

Figure 2 and S. Figure 2, could have been more informative if the authors had included CD235a to quantify the frequency of CD71+CD235a+ T cells. Although CD71 is included, its unclear whether those cells are CECs or not. I believe those can be any activated cells considering that CD71 gets upregulated on activated myeloid/lymphoid cells.

Figure 3, implies that GEMM and BFU-E/CFU-E are increased in the conditioning culture media;

therefore, these observations support my previous comment above that SARS-CoV-2 modulated hematopoiesis but not erythropoiesis.

The conclusion in lines 402-403 might not be correct. You can not rule out stress hematopoiesis by measuring a few cytokines. Did the author quantify the levels of GMC-SF or other stimulating factors that could enhance myelo-erythropoiesis?

The authors should not make any conclusions when there is no significant difference between the groups. This is commonly observed throughout the manuscript. For example, they have claimed that HIF-1a is upregulated in COVID-19+ but Fig. 5c shows no significant difference.

Any attempt was made to quantify the levels of maternal cytokines and maternal CECs to connect the dots?

I suggest authors should prevent overstating their findings since the nature of their work is highly descriptive and no convincing mechanistic data is provided to support some of their claims.

The study limitations should be included (e.g. sample size, not quantifying CECs in fresh samples). The storage issue; should be stored in Trizol right after collection.....

SARS-CoV-2 variants may have differential effects on hematopoiesis and the erythroid signature, hence, authors may discuss this (PMID: 35943266).

Finally, to be consistent with the literature, the authors are encouraged to use the term (CD71+ erythroid cells) if possible.

Response to Referees

We acknowledge the constructive comments of the Referees and have taken into account all their remarks. Please find below our point-to-point response and our revised manuscript.

Reviewer #1 (Remarks to the Author):

In this paper Alkobtawi et al. report on their study on fetal response to maternal SARS-CoV-2 infection during pregnancy. The authors investigated the transcriptome and phenotype of cord blood cells in a retrospective single-center cohort study and report an erythroid cell transcriptional signature in cord blood from mothers with symptomatic SARS-CoV-2 infection, as well as an increased erythroid differentiation of fetal hematopoietic multipotent progenitors in quality and quantity.

The authors interpret these findings as possible fetal stress erythropoiesis induced by maternal hypoxia. The study is novel and relevant to the field, as the conclusions would add to existing reports of potential placental problems in SARS-CoV-2 infection.

Overall, the reported claims can be made from the existing transcriptomics, clonogenicity and FACS analysis, and especially the latter is comprehensively performed and described.

We thank Reviewer 1 for evaluating our work and providing us with helpful comments.

However, there are some issues:

1. The title “Enhanced fetal erythropoiesis in response to symptomatic COVID-19 infection during pregnancy” is not differentiating between disease and virus. COVID-19 is the disease, SARS-CoV-2 the virus. There is no COVID-19 infection. Therefore, the title should be corrected, e.g., to „Enhanced fetal erythropoiesis in response to COVID-19 during pregnancy“ or „Enhanced fetal erythropoiesis in response to symptomatic SARS-CoV-2 infection during pregnancy“

We apologize for this incorrect terminology. We have now changed the title to “Enhanced fetal myelopoiesis in response to symptomatic SARS-CoV-2 infection during pregnancy”.

2. Because no reviewer token is provided, the generated RNA-seq data (GSE224063) cannot be evaluated.

We deeply apologize for this oversight. The token to access the RNAseq dataset GSE224063 is “yjgzwygygvnctjen”.

3. According to the author’s description, the cord blood was already centrifuged at 900 g for 12 min before Ficoll-Paque gradient centrifugation. Is this a misunderstanding? Plasma could have been easily collected after Ficoll-Paque PLUS gradient centrifugation.

As noticed by Reviewer 1, we performed an initial centrifugation step at 900 g for 12 min prior to Ficoll-Paque gradient centrifugation because we did not want to dilute the plasma samples (to be able to quantify even small amounts of cytokines). Indeed, it is recommended to dilute samples in PBS (up to 50 mL) before proceeding with Ficoll-Paque gradient centrifugation, so we were concerned that it would be difficult to accurately quantify overdiluted plasma samples. We have clarified this point in the Methods.

4. Usually gradient centrifugation by Ficoll-Paque is done for more than 10 min (30-40 min). Was there a specific reason for using only 10 min. A reference for a 10 min protocol is recommended. Was 10 min always employed? Blood from infants after birth may contain early erythroid cells (CD71+, who show a high TAL1 expression, Correia et al. *Leukemia* 30, 1968–1978 (2016)), that may contaminate the upper buffy coat layer (<https://www.urmc.rochester.edu/medialibraries/urmcmedia/rochester-human-immunology-center/documents/HIC-1-0020Approved.pdf>). Therefore, the question of a possible contamination and pre-analytic error because of rather short centrifugation time would have to be asked. However, if the isolation of mononuclear cells from cord blood was always done in the same way, this would be a balanced error.

We used Blood-Sep-Filter Tube (Dominique Dutscher France, Ref #016760) to perform Ficoll-Paque gradient centrifugation. Therefore, we centrifuged during 10 min at 1000 g following the manufacturer's instructions (https://pdf.dutscher.com/doc/016780/016780_MEen.pdf). Of note, we corrected the speed of this centrifugation as there was a spelling mistake in the previous manuscript. In addition, we confirm that the isolation of mononuclear cells from cord blood was always done in the same way for all our samples.

5. RNA extraction and sequencing should include some more information, e.g., was the total RNA depleted for ribosomal RNA (likely?). Otherwise, there may not have been the sequencing depth to detect more differences between the groups.

We apologize for this lack of information. We have now included more information about RNA extraction and sequencing in the Methods section. Total RNA were not depleted for ribosomal RNA, however we purified poly-A containing mRNA molecules by using oligodT beads. In addition, we now provide in the new Supplementary Table 1 the sequencing depth of all the samples (mean sequencing depth of 39 millions of read per sample, minimum of 28 millions of reads).

6. Also, the protocol/kit used for RNA-seq library preparation should be named.

Again, we apologize for forgetting to mention this point. The kit we used was the NEBNext Ultra II RNA Library Prep Kit from New England BioLabs. This information has now been added to the RNA extraction and sequencing section of the Methods.

7. Why was an outdated version of STAR (2.7.6a) employed (current as of 2022: 2.7.10b, <https://github.com/alexdobin/STAR/blob/master/CHANGES.md>)

At the date we started the analyses of our samples (end of 2020), the updated version of STAR was the 2.7.6a that we kept for the entire project.

8. Why was an outdated version of R (3.6.1) employed (current as of 2022: 4.2.2. <https://stat.ethz.ch/pipermail/r-announce/2022/000680.html>)

We used R version 3.6.3 (and not 3.6.1). This was the version we were using at the date we started the analyses of our samples in 2020. Therefore, we kept this version for the entire project.

9. Which log2FC cut-off was employed before doing downstream analysis (esp. Ingenuity Pathway Analysis)?

Analyses using Ingenuity Pathway Analysis (IPA) (Qiagen) were performed for all transcripts that were differentially regulated with an adjusted p-value threshold of 0.05. We did not perform any selection based on the log2FC since we assumed that even a small change of transcript expression may be biologically meaningful between 2 conditions (and therefore statistically significant). This has now been better explained in the Methods section.

10. The authors have made many sophisticated comparisons with existing transcriptome data. For ease of reproduction, the relevant module gene sets should be presented in a supplementary table.

Module gene sets used for Figure 1c from Zheng et al Mol Syst Biol 2018 are presented in the new Supplementary Table 3.

Module gene sets used for Supplementary Figure 3 from Bernardes et al Immunity 2020 are presented in the new Supplementary Table 4.

11. The finding that transcriptomes from male and female cord bloods cluster separately (Fig. 1a) is not discussed. While adjustment for gender in SARS-CoV-2 infection is justified a short discussion, taking into account the existing literature is recommended.

We apologize for not discussing about this finding. We have now included a paragraph in the Discussion to comment on the literature on this subject.

Reviewer #2 (Remarks to the Author):

The manuscript by Alkobtawi et al. claims that SARS-CoV-2 infection in pregnant women results in enhanced erythropoiesis. This is in line with previous reports showing the expansion of erythroid precursors/progenitors in the peripheral blood of COVID-19 patients.

The descriptive nature of the study is the major limitation. Small sample size and limited mechanistic studies to support their findings.

We thank Reviewer 2 for evaluating our work and providing us with helpful comments. We have now detailed the limitations of our study, including its descriptive nature, the small size of our cohort and the difficulty to perform elaborate mechanistic studies in the Discussion section.

It is well documented that cord blood is enriched with erythroid precursors/progenitors defined as CD71+ erythroid cells (CECs) (e.g. PMID: 30272151, PMID: 29735482, and PMID: 24196717). Under physiological conditions, the buffy coat (PBMCs) from the cord blood contains 50-70% CECs. So, it would have been highly informative to provide flow plots of the frequency of CECs in different groups. There is a major concern in the methodology. CECs from cord blood or any other sources do not resist the freeze-and-thaw process. Basically, they get lysed in the freezing media unless they have a hybrid erythroid/myeloid phenotype. Based on the described methodology that the authors have stored the cells at -80 until use. The integrity of total RNA from CECs is questionable. Because the cells has to be washed off the freezing media prior to RNA isolation. Hence, all the content of CECs will be washed away.

We thank Reviewer 2 for pointing out these papers about CD71+ erythroid cells. In this work, we aimed to investigate the phenotype of cord blood immature hematopoietic cells upon maternal COVID-19 (that are not sensitive to freezing). Therefore, analyses performed for Figures 2 and 3 were done in the CD34+ population.

Of note, we harvested cord blood samples at any time of the day or night and immediately processed them to guaranty the quality of our samples (maximum delay between delivery and harvest was 1 hour). Therefore, we had to freeze the cells for downstream analyses, such as FACS and methylcellulose cultures, as it will not have been feasible to immediately analyze them.

About the methodology, we apologize for our lack of clarity. Indeed, the cord blood mononuclear cells used to perform RNAseq were never frozen. In brief, at the end of the isolation process, cells were divided into 3 tubes: 1) immediate lysis of fresh cells for total RNA extraction; 2) immediate freezing at -80°C (without freezing media) for total protein extraction; and 3) immediate freezing in freezing media for cell phenotyping analyses and culture.

Therefore, we can assume that CEC were indeed present in the cells analyzed by RNAseq. However, they were not analyzed in Figures 2 and 3 (in which cells had previously been frozen). Since, as you mentioned, there is a risk of losing them due to freezing, we cannot confidently measure them in our samples. Nonetheless, and to answer your question, we re-analyzed in our FACS data the potential population of CEC expressing CD71 and CD36 (we do not have CD235a marker as published in the original description). We found very low amount of CEC confirming your point about the vulnerability of these cells to freezing. However, there was a tendency of having more CEC in COVID+S samples, but this was not significant (see Figure below). Therefore, we believe we cannot conclude definitely about CEC in our samples (and thus will not show the results below), but we believe that they should be increased in our COVID+S since erythroid-committed CD34+ progenitors are increased in these samples. We have now added a paragraph in the Discussion to discuss about CEC and the limits of our work about these cells.

Figure: (A) Gating strategy for the identification of CD71+ CD36+ cells (CD71+ erythroid cells) in representative COVID- and COVID+S samples. (B) Quantification of CD71+ CD36+ cells as a percentage of live cells in COVID- (n=3) and severe COVID+S (n=3) cord blood mononuclear cells. Data are presented as means with SEM and individual values. Statistical analyses were performed with Mann-Whitney test. (ns) not significant.

Since erythroid precursors have a very low RNA content, I speculate that samples from COVID-19+ might have had a greater number of erythroid progenitors or nucleated CECs. I think it's impossible to compare CD45+ among the erythroid signature in the bulk seq to address this question. It will be feasible in scRNASeq.

As indicated above, we are not able to measure with certainty the levels of CEC in our samples. But we believe that there should be more CEC in COVID+S samples (as also suggested by the slight increase of CD71+ CD36+ cells in COVID+S).

We were not able to compare CD45+ among the erythroid signature in the RNAseq data. However, CD45 expression was not different between COVID-/A and COVID+S (log2FC -0.039, adjusted p-value 0.897).

Unfortunately, we did not perform scRNAseq on our samples and we did not store cells for this purpose.

Regardless of the quality of RNA, there is a possibility that COVID-19 modulates hematopoiesis rather than erythropoiesis. The text/title should be modified accordingly. By any chance did the author compare CD45+ subpopulations among CECs between COVID-19+/- groups by flow cytometry?

We thank the reviewer for this comment. We agree that the increase of CFU-GEMM in methylcellulose cultures and of MPP F1 by FACS suggests a broader modulation of myelopoiesis rather than just erythropoiesis. We have therefore modified the title and the text accordingly.

In addition, we have tried to measure CD34+ and CD45+ cells in the CD71+ CD36+ CEC population as shown below in a representative COVID+S sample, but the proportion of these cells was extremely low. Given the bias about CEC measurement in our samples, we therefore do not think these results are reliable.

I did not see the RIN score for the analyzed samples.

We apologize for this oversight. The RIN scores for all the samples are now provided in Supplementary Table 1.

Figure 2 and S. Figure 2, could have been more informative if the authors had included CD235a to quantify the frequency of CD71+CD235a+ T cells. Although CD71 is included, its unclear whether those cells are CECs or not. I believe those can be any activated cells considering that CD71 gets upregulated on activated myeloid/lymphoid cells.

As previously explained, our experimental design was not optimal to measure CEC since cord blood mononucleated cells were analyzed by FACS after freezing. However, we did include CD36 in our panel that enabled us to measure CD71+ CD36+ cells which we think are similar to CD71+ CD235a+.

Of note, we analyzed in Figure 2 and Supplementary Figure 2 CD71+ cells among the proportion of CD34+ cells. Therefore, as suggested by the reviewer, we believe these CD71+ cells are not CEC but rather hematopoietic stem cells/progenitors committing to erythroid differentiation. This is now discussed in the revised manuscript.

Figure 3, implies that GEMM and BFU-E/CFU-E are increased in the conditioning culture media; therefore, these observations support my previous comment above that SARS-CoV-2 modulated hematopoiesis but not erythropoiesis.

We thank you for this observation as it is an important finding to discuss. Thus, we have now modified the text accordingly.

The conclusion in lines 402-403 might not be correct. You can not rule out stress hematopoiesis by measuring a few cytokines. Did the author quantify the levels of GMC-SF or other stimulating factors that could enhance myelo-erythropoiesis?

As suggested by the reviewer, we have now performed a quantification of GM-CSF (new Figure 4i). We observed no difference between COVID+S and COVID-/A samples.

However, we agree with Reviewer 2 that we have not performed an exhaustive quantification of all the cytokines that could enhance myelo-erythropoiesis, which is why we have toned down our statement in the text.

The authors should not make any conclusions when there is no significant difference between the groups. This is commonly observed throughout the manuscript. For example, they have claimed that HIF-1a is upregulated in COVID-19+ but Fig. 5c shows no significant difference.

We deeply apologize for this. We have critically reviewed our manuscript and toned down our statements in the text when necessary.

About the Figure 5c, we agree that there is only a trend of increase, however we believe the decrease of VHL reflects HIF-1 α pathway activation.

Any attempt was made to quantify the levels of maternal cytokines and maternal CECs to connect the dots?

Unfortunately, we did not have access to any maternal samples. This is a limit to our study that we have now clearly explained in the new Study limitations paragraph in the Discussion.

I suggest authors should prevent overstating their findings since the nature of their work is highly descriptive and no convincing mechanistic data is provided to support some of their claims.

We have now toned down some of our statements regarding our mechanistic hypotheses.

The study limitations should be included (e.g. sample size, not quantifying CECs in fresh samples). The storage issue; should be stored in Trizol right after collection.....

SARS-CoV-2 variants may have differential effects on hematopoiesis and the erythroid signature, hence, authors may discuss this (PMID: 35943266). Finally, to be consistent with the literature, the authors are encouraged to use the term (CD71+ erythroid cells) if possible.

We have now included a Study limitations paragraph at the end of the Discussion. We have also discussed how SARS-CoV-2 variants may have differential effects on erythropoiesis and the erythroid signature. And finally, we are using the term CD71+ erythroid cells to clearly designate these cells.

REVIEWERS' COMMENTS:

Reviewer #1 (Remarks to the Author):

For the summary I refer to my earlier review of Alkobtawi et al.

The authors have done a sufficient revision.

My objections regarding methodology, data, missing discussion points and terminology have been adequately addressed.

Reviewer #2 (Remarks to the Author):

The authors have addressed most of my comments and overall their manuscript has benefited from these improvements. However, my major concern is that the title should be changed to hematopoiesis not myelopoiesis. I had suggested using hematopoiesis but the authors replaced the "erythropoiesis" with "myelopoiesis". There is no clear evidence in the manuscript that SARS-CoV-2 drives or skews erythropoiesis to myelopoiesis but instead their main finding supports erythropoiesis. There is a small evidence for myelopoiesis. Therefore, "myelopoiesis in the title should be either changed to hematopoiesis, which is a general and broader term. Alternatively, replacing the myelopoiesis in the title to erythropoiesis/myelopoiesis. According to the title the text should be modified.

Unfortunately, changes were not highlighted throughout the manuscript, which should have been.

Response to Referees

Reviewer #1:

For the summary I refer to my earlier review of Alkobtawi et al.

The authors have done a sufficient revision.

My objections regarding methodology, data, missing discussion points and terminology have been adequately addressed.

We thank Reviewer 1 for reviewing our work and helping us improve our manuscript.

Reviewer #2:

The authors have addressed most of my comments and overall their manuscript has benefited from these improvements. However, my major concern is that the title should be changed to hematopoiesis not myelopoiesis. I had suggested using hematopoiesis but the authors replaced the "erythropoiesis" with "myelopoiesis". There is no clear evidence in the manuscript that SARS-CoV-2 drives or skews erythropoiesis to myelopoiesis but instead their main finding supports erythropoiesis. There is a small evidence for myelopoiesis. Therefore, "myelopoiesis in the title should be either changed to hematopoiesis, which is a general and broader term. Alternatively, replacing the myelopoiesis in the title to erythropoiesis/myelopoiesis. According to the title the text should be modified.

We thank Reviewer 2 for reviewing our work and helping us improve our manuscript.

We have now changed the term "myelopoiesis" to "hematopoiesis" in the text and in the title.

Unfortunately, changes were not highlighted throughout the manuscript, which should have been.

We apologize as it seems that you were not able to see the changes we have made to the text (we used the track changes function of Word). We have now highlighted in yellow these changes, which are also visible as "track changes".